# In-Situ Characterisation of Charge Transport in Organic Light-Emitting Diode by Impedance Spectroscopy

**Pavel Chulkin**

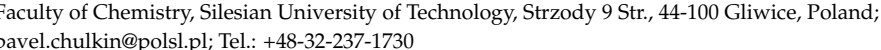

Faculty of Chemistry, Silesian University of Technology, Strzody 9 Str., 44-100 Gliwice, Poland;
pavel.chulkin@polsl.pl; Tel.: +48-32-237-1730

**Abstract:** The article demonstrates an original, non-destructive technique that could be used to in situ monitor charge transport in organic light-emitting diodes. Impedance spectroscopy was successfully applied to determine an OLED's charge carrier mobility and average charge density in the hole- and electron-transport layer in a range of applied voltages. The fabricated devices were composed of two commercially available materials: NPB (N,N′-di(1-naphthyl)-N,N′-diphenyl-(1,1′-biphenyl)-4,4′-diamine) and TPBi (2,2′,2′′-(1,3,5-Benzinetriyl)-tris(1-phenyl-1-H-benzimidazole)) as hole- and electron-transport layers, respectively. By varying the thicknesses of the hole-transport layer (HTL) and the electron-transport layer (ETL), correlations between layer thickness and both charge carrier mobility and charge density were observed. A possibility of using the revealed dependencies to predict diode current–voltage characteristics in a wide range of applied voltage has been demonstrated. The technique based on a detailed analysis of charge carrier mobilities and densities is useful for choosing the appropriate transport layer thicknesses based on an investigation of a reference set of samples. An important feature of the work is its impact on the development of fundamental research methods that involve AC frequency response analysis by providing essential methodology on data processing.

**Keywords:** organic light-emitting diode; impedance spectroscopy; mobility; charge density; transport layers





## 1. Introduction

Organic light-emitting diodes (OLEDs) are an object of active research in modern material science. Being an effective source of light, they provide an energetic application of perspective organic electro-luminescent and electro-conductive materials [1–5]. The first OLEDs employed conductive polymers as active components of the emission layer, providing a useful application of this exceptional kind of material [1,3]. During the last decade, researchers' interests have shifted to small molecules. The concept covers non-polymer organic materials that form a conductive layer in the solid-state. The replacement of polymers by molecules provides a broader range of ways for molecule design as well as the facilitation of control of the deposition process. The diversity of organic compounds (along with their perspective utilization in organic electronics) became a motivating factor for the development of organic synthetic methodology [6,7]. One more advantage is that molecular compounds can be easily studied in solutions via electrochemical and spectroscopic techniques, before the preparation of a solid film [8–10]. An OLED consists of several layers, the major one being the recombination layer (or interface), where two types of charge carriers meet and recombine. The transport layers serve for delivery of charge carriers from the outer electrodes to the recombination interface. Though the recombination layer is crucial for device emission, one has to take into account the transport layer conductivity that assures a lossless charge delivery. The ideal transport layer must be able to transfer as many charge carriers as possible as fast as possible at as low as the possible applied voltage.

In the case of semiconducting materials, especially organic semiconductors, conductivity is not a constant parameter that can be attributed to the individual material. It is a product of charge density ($n$) and charge mobility ($\mu$). Those two mentioned parameters are functions of the electric field and depend on layer thickness [11]. Charge carrier mobility is estimated for individual materials using time-of-flight (TOF) and thin-film transistor (TFT) methods [12–16]. Several other attempts were also made to estimate charge carrier mobility under space charge limited current conditions or using frequency analysis of AC response of organic thin films between indium tin oxide (ITO) and metal as electrical contacts [17,18]. Though the proposed approach is well substantiated, it is based on a theoretical background derived for inorganic semiconductors that requires modification to describe the phenomena inherent in organic materials (e.g., absence of charged donor or acceptor in a neutral state or strong dependence of the concentration of charge carriers on the electric field).

It is worth mentioning that not only charge carrier mobility ($\mu$) but a complex of three parameters ($n$, $\mu$, $E$) has to be considered to predict the full conductivity characteristics of a material. One must know not only how fast charge carriers are moving but also how many of them are there and how large a driving force is required to run them. A mathematical simulation provides an attractive and easy way to look at a range of materials to select the best option based on the result of the virtual measurement. Multilayer OLEDs were firstly thoroughly described in different works [19,20]. The simulation was based on a numerical solution of a system of derivative equations corresponding to the migration and diffusion of charge carriers in all layers. In later works, the authors of the mentioned articles addressed more complex effects such as localized states [21] and shallow traps [21,22].

Despite the unlimited possibilities of theoretical modulation, we devoted this work to the analysis of experimental data. The use of mathematical simulations was minimized to avoid the influence of theoretical expectations on the obtained results and develop a characterization technique that can be used to observe charge transport in OLED in real-time mode.

Several assumptions were made; nevertheless, the model was investigated to prove our theoretical approach. First, the migration flux of charge carriers caused by a potential gradient was regarded as dominant in comparison to the diffusion flux. Therefore, data analysis was carried out in the range of high applied voltage: from 5 to 10 V. Second, only one type of charge carriers in a single material was taken into account, i.e., hole transport layer (HTL) materials conducted holes only while electron transport layer (ETL) materials conducted electrons. Such an assumption was widely used in previous works concerning the conductivity of semiconductors [23,24].

Impedance spectroscopy is the best method to characterize conductive and capacitive properties by applying the AC frequency signal and its analysis. Two types of charge carriers (holes and electrons) have different mobility [25–27]. Thus, the inequality of complex AC frequency responses of HTL and ETL enables separate consideration and analysis of resistance and capacitance of each layer. Recently we worked out an impedance spectroscopy-based technique that could be used to estimate values of charge carrier density and mobility within the OLED device during operation [28]. The objective of the current work is to observe the effect of transport layer thickness and develop an approach to predict material conductivity properties and control them by the thickness of the device's internal layers.

## 2. Materials and Methods

Five types of devices having the same qualitative composition based on active compounds NPB (HTL) and TPBi (ETL) were chosen to demonstrate the method and elucidate an effect of layer thickness (Figure 1). Their compositions and designations are shown below.

**L$_H$M$_E$**—ITO | NPB (60 nm) | NPB:TPBi (1:1) (20 nm) | TPBi (50 nm) | LiF (1 nm) | Al
**S$_H$M$_E$**—ITO | NPB (20 nm) | NPB:TPBi (1:1) (20 nm) | TPBi (50 nm) | LiF (1 nm) | Al

**M$_H$M$_E$**—ITO | NPB (40 nm) | NPB:TPBi (1:1) (20 nm) | TPBi (50 nm) | LiF (1 nm) | Al
**M$_H$S$_E$**—ITO | NPB (40 nm) | NPB:TPBi (1:1) (20 nm) | TPBi (30 nm) | LiF (1 nm) | Al
**M$_H$L$_E$**—ITO | NPB (40 nm) | NPB:TPBi (1:1) (20 nm) | TPBi (70 nm) | LiF (1 nm) | Al

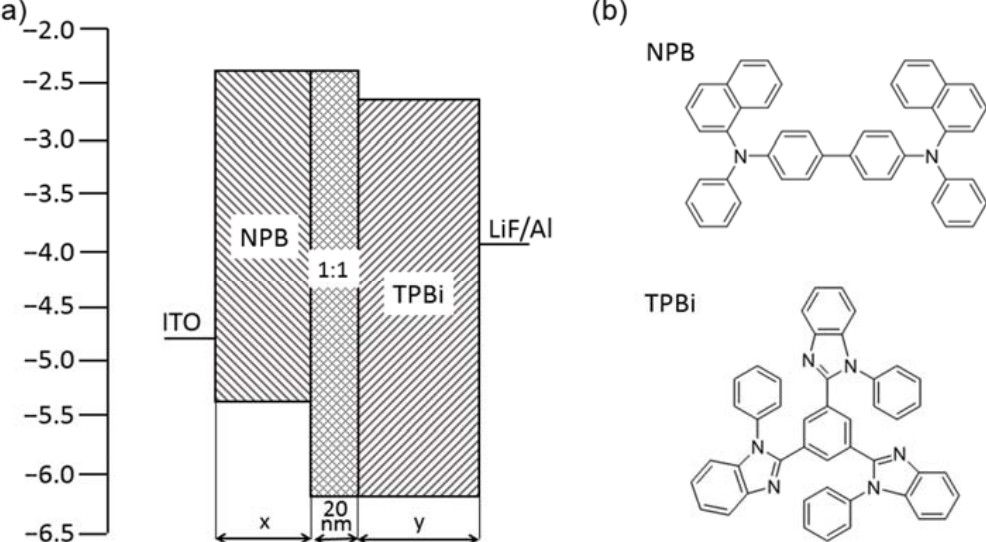

**Figure 1.** Energy diagrams of the device under investigation (**a**) and the molecular structures of the hole transport material NPB and an electron transport material TPBi (**b**). The variables x and y designate the thickness of the transport layers that we varied between different samples.

Three variants of layer thickness are assigned by letters: S—small, M—medium, and L—large. The first letter corresponds to the HTL thickness (lower index H) and the second corresponds to the ETL thickness (lower index E). The devices of that type were reported as efficient OLEDs employing simple architectures involving two organic compounds, forming an exciplex [29,30].

OLED devices were fabricated using pre-cleaned ITO-coated glass substrates purchased from Visiontek with a sheet resistance of 15 $\Omega$ cm$^{-2}$ and an ITO thickness of 150 nm. They were patterned so that the OLED devices had two pixels sized 4 $\times$ 4 mm$^2$ and two pixels sized 4 $\times$ 2 mm$^2$. The small molecule layers and the cathode (Al) layer were thermally evaporated using the Kurt J. LeskerSpectros II at 5 $\times$ 10$^{-7}$ mbar pressure. The materials were deposited at rates from 0.5 to 1 Å s$^{-1}$. Built-in, pre-calibrated quartz-crystal microbalance sensors inside the evaporation chamber controlled the thickness. At least three diodes of each type, each including four pixels, have been fabricated to control the reproducibility of the results.

The characterization of the OLEDs emission was conducted in a 10 inches integrating sphere (Labsphere) connected to a Source Meter Unit and calibrated with an NIST calibration lamp. Electrochemical measurements were carried out using BioLogic SP-300 Potentiostat. Impedance spectra were obtained in a 1 MHz–1 Hz frequency range with 20 points per decade in a logarithmic scale (the total number of frequencies in one spectrum was 121). The angular frequency ($\omega$) used for mathematic data treatment is 2$\pi$ times larger than the frequency ($f$). AC voltage amplitude was 10 mV. The voltage was changed in a range from 0 V to 10 V incremented by 0.1 V or 0.25 V in staircase mode, before each step the voltage was maintained for 20 s to achieve stationary current conditions for the spectrum recording. Analysis of electrochemical impedance spectra and determination of equivalent circuit parameters was accomplished using the EIS analyser program [31].

# 3. Results

## 3.1. OLEDs Characterization by Impedance Spectroscopy

An example of an impedance spectra of a reference OLED and its evolution with applied voltage is presented in Figure 2. The Bode plot (Figure 2a) is more useful than the complex plane (Nyquist) plot (Figure 2b) when comparing spectra at different applied voltages, since the difference in impedance values is huge.

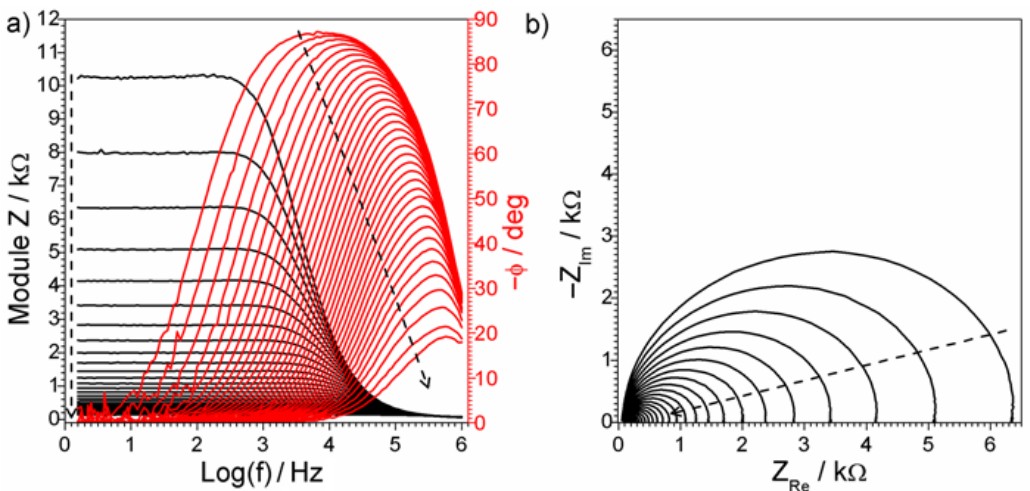

**Figure 2.** The $M_H M_E$ OLED impedance spectra (typical of OLED) at a different applied voltage from 5 to 10 V: Bode plot (**a**) comprising impedance module (black) and phase shift (red) frequency dependence plots, Nyquist plot (**b**). Dashed arrows indicate the evolution of spectra with voltage increase from 0 V to 10 V, with an increment of 0.2 V.

Though an effective capacitance is often used to characterize OLED electric properties [17,18,32], here we intend to promote data treatment in terms of impedance. The reason for this is that, under operating conditions, the device is conductive and the apparent capacitance is a parameter of dimension $\Omega^{-1} \cdot Hz^{-1}$ (equivalent to Farad), arising from the complex admittance. The impedance-featuring approach allows for the easy implementation of the theoretical apparatus developed for charge transport phenomena and to avoid a conservative treatment of a diode as a shunted capacitor.

Both plots in Figure 2 express the typical decrease of an impedance with voltage. At low voltages, the current is very small so the device behaves as a capacitor (like a dielectric material placed between two metal contacts). An increase in voltage is accompanied by an increase of conductivity, which results in the observed tendency in capacitive behaviour. At high voltage, a non-zero phase shift remains only in the range of high frequencies (Figure 2a), where capacitive impact prevails.

An equivalent electrical circuit (Figure 3a) was found to fit the experimental spectrum when the current achieved considerable values and the device started emitting.

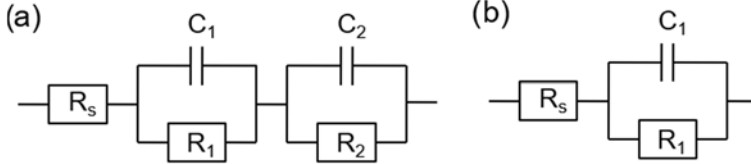

**Figure 3.** The equivalent electrical circuits that could correspond to a semicircle and quasi-semicircle complex plane impedance plot. Text designations: R-(R/C)-(R/C) (**a**), R-(R/C) (**b**).

The five-element equivalent circuit was found to satisfactorily fit the experimental data. The complete mathematical discussion justifying the reasonability of using this model is included in Appendix B. Nevertheless, the circuit could precisely fit the experimental

spectrum only in the range of high voltages when the current value of about 0.1 mA cm$^{-2}$ was achieved. Results obtained at lower current conditions could not be successfully processed due to the following reasons: a very high resistance (more than 10$^7$ $\Omega$) and the signal noise (wide distribution of measurement points), probably caused by the absence of a stable conductive path within the organic material.

Another methodological issue concerned with the equivalent circuit is worth mentioning. It is believed that the impedance spectrum corresponding to the equivalent circuit shown in Figure 3a would have a form of two semicircles, whereas one semicircle would correspond to a simpler circuit (Figure 3b) consisting of three elements: $R_s$ and $R_1$ and $C_1$ (without $R_2$ and $C_2$), connected in a parallel manner. Two distinct semicircles would be observed in the case of a considerably high difference of products $R_1 \cdot C_1$ and $R_2 \cdot C_2$. When the difference is about one order, two semicircles overlap to form one arch, which could be mistakenly analysed by using a simpler model (Figure 4a). The mistake could likely be made when an OLED investigation is performed. The resistance and capacitance of two organic materials do not differ significantly, so precise analysis in a wide range of frequencies has to be accomplished to extract all necessary equivalent circuit parameters. Figure 4a demonstrates the correct fitting of the experimental spectrum of OLED by the five-element circuit and three erroneous fitting results by using the three-element equivalent circuit. The wrong results could be regarded as correct if a narrow range of frequencies is considered. A set of values of two capacitance and resistance values as functions of the applied voltage is presented in Figures 4b and A1 (Appendix A). Each point of the graphs was obtained through analysis of individual spectrum registered at the corresponding voltage. To calculate the values, a special software was used [31].

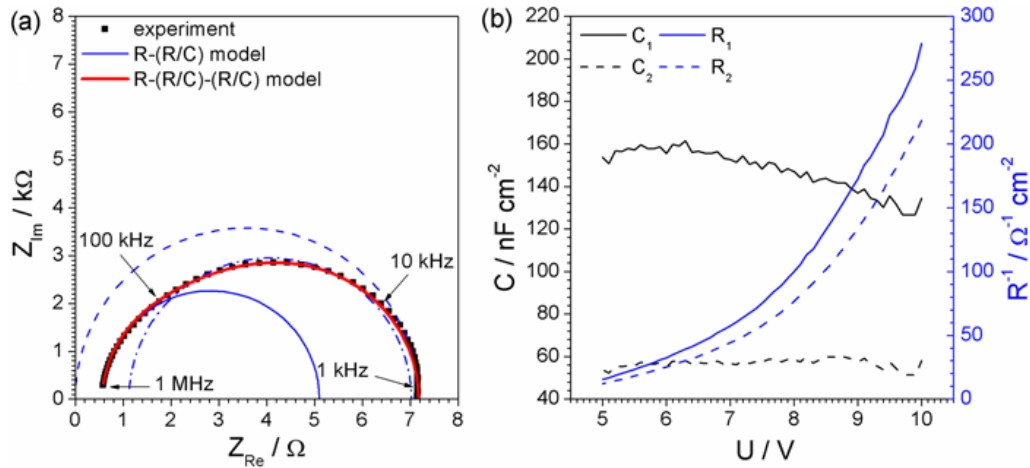

**Figure 4.** (**a**) Comparison of correct fitting of experimental data with five-element (red line) equivalent electric circuits and erroneous fitting with three-element (blue lines) within the narrow frequency ranges 1–100 kHz, 100–10 kHz, 10–1 kHz; (**b**) equivalent electrical circuit parameters $C_1$, $C_2$, $R_1^{-1}$, $R_2^{-1}$ of $\mathbf{M_H L_E}$ device as functions of the applied voltage.

The equivalent circuit (or model-fitting approach) has a cogent advantage—it enables estimation of the parameters when only a particular, relatively narrow range of the frequency points is at the disposal. It helps to solve the problem with the inability of the precise registration of the low frequency ($\omega \to 0$) and high frequency ($\omega \to \infty$) points.

According to Figure 4b, the capacitances do not change significantly in the operating voltage range. The variation of the values did not exceed 30%. Both inverse resistance (conductivity) voltage dependence plots (Figure 4b) otherwise show clear tendencies. The increase of conductivity with voltage is observed due to the gain in the charge carriers' injection. The conductance–voltage dependence plots are smooth and are not reliable to signal noise, as in the case of capacitance. The acquisition of capacitance and resistance

plots is a crucial step in the OLED analysis procedure. However, having those dependencies is still not enough to consider the fundamental physical values of charge carriers.

A special theoretical model had to be worked out to estimate the values of charge density and mobility from the values of parameters determined by the impedance spectroscopy method. We used a classical impedance function derivation strategy, which includes the following stages: (i) analysis of time-dependent parameters that affect current and voltage; (ii) representation of current oscillation ($\Delta i$) as the Taylor series; (iii) expressing partial derivatives and phasors of oscillating parameters through known values and phasors of voltage ($U$) or current ($i$) [33]. The final expression containing only values of $U$ and $i$ is rearranged to eliminate their ratio and obtain the final formula for complex system impedance.

A detailed single-stepping mathematical procedure concerning the derivation of the working formulas based on charge transport in organic layer (Figure A4) and equivalent electrical circuit model (Figure A5) is presented in Appendix B. The final expressions relating experimentally determined values ($i$, $R$, $C$) with physical parameters ($n$, $E$, $\mu$) are given below (1)–(3):

$$\mu = \frac{d^2}{iR^2C} \frac{1}{\left(1 + \sqrt{\frac{dC}{\varepsilon\varepsilon_0}}\right)^2} \sqrt{\frac{dC}{\varepsilon\varepsilon_0}} \tag{1}$$

$$n = \frac{iRC}{zed}\left(1 + \sqrt{\frac{\varepsilon\varepsilon_0}{dC}}\right) \tag{2}$$

$$E = \frac{iR}{d}\left(1 + \sqrt{\frac{dC}{\varepsilon\varepsilon_0}}\right) \tag{3}$$

where $n$ is charge carrier density; $i$—current density; $R$—resistance (normalized by surface area); $z$—unit charge of a carrier (accepted to be 1); $e$—elementary charge; $\varepsilon$—relative material permittivity; $\varepsilon_0$—vacuum permittivity; $d$—thickness of the individual material layer (i.e., HTL or ETL); $C$—capacitance (normalized by surface area); $E$—electric field intensity; and $\mu$—charge mobility.

The formulas are eligible for each internal layer if its capacitance and resistance are known. Hereinafter, the set of parameters corresponding to the hole-transport layer are denoted with the lower index "h", while the parameters attributed to the electron-transport layer are denoted with the lower index "e". The third input parameter (the current) is equal for all device layers, as they are connected in series. The evaluated characteristics appeared to be very useful to trace tendencies between conductivity properties and layer thickness.

### 3.2. Experimental Data Processing and Discussion

All the results obtained from impedance spectra were processed according to Formulas (1)–(3). The obtained voltage dependencies of charge carrier mobility, density, and electric field are presented in Figure 5: the left column corresponds to the HTL (Figure 5a,c,e) and the right corresponds to the ETL (Figure 5b,d,f) (e.g., HTL thickness was different within the number of devices $S_H M_E$ (small 20 nm), $M_H M_E$ (medium 40 nm), and $L_H M_E$ (large 60 nm), so the curves corresponding to these devices are shown in the left column of figures (Figure 5a,c,e)). The right column of figures depicts the tendencies related to ETL thickness change in a row $M_H S_E$, $M_H M_E$, and $M_H L_E$.

As Figure 5 shows, both $\mu$ and $n$ increase with voltage. Such behaviour of mobility was already observed [34] and is predicted theoretically by the Poole–Frenkel equation, which will be discussed below. The increase of charge density with voltage is obvious because a higher electric field causes the injection of more charge carriers (Figure 5c,d).

Figure 5c,d show a correlation between layer thickness and charge carrier density. The lowest density at the same voltage applied was observed in the case of the thickest layer (devices $L_H M_E$ and $M_H L_E$), while densities in the thinnest layer (devices $S_H M_E$ and $M_H S_E$) were the highest. The effect might be explained in terms of charge injection by an

electric field. The electric field in a thin layer is higher than that in a thick layer at the same applied voltage.

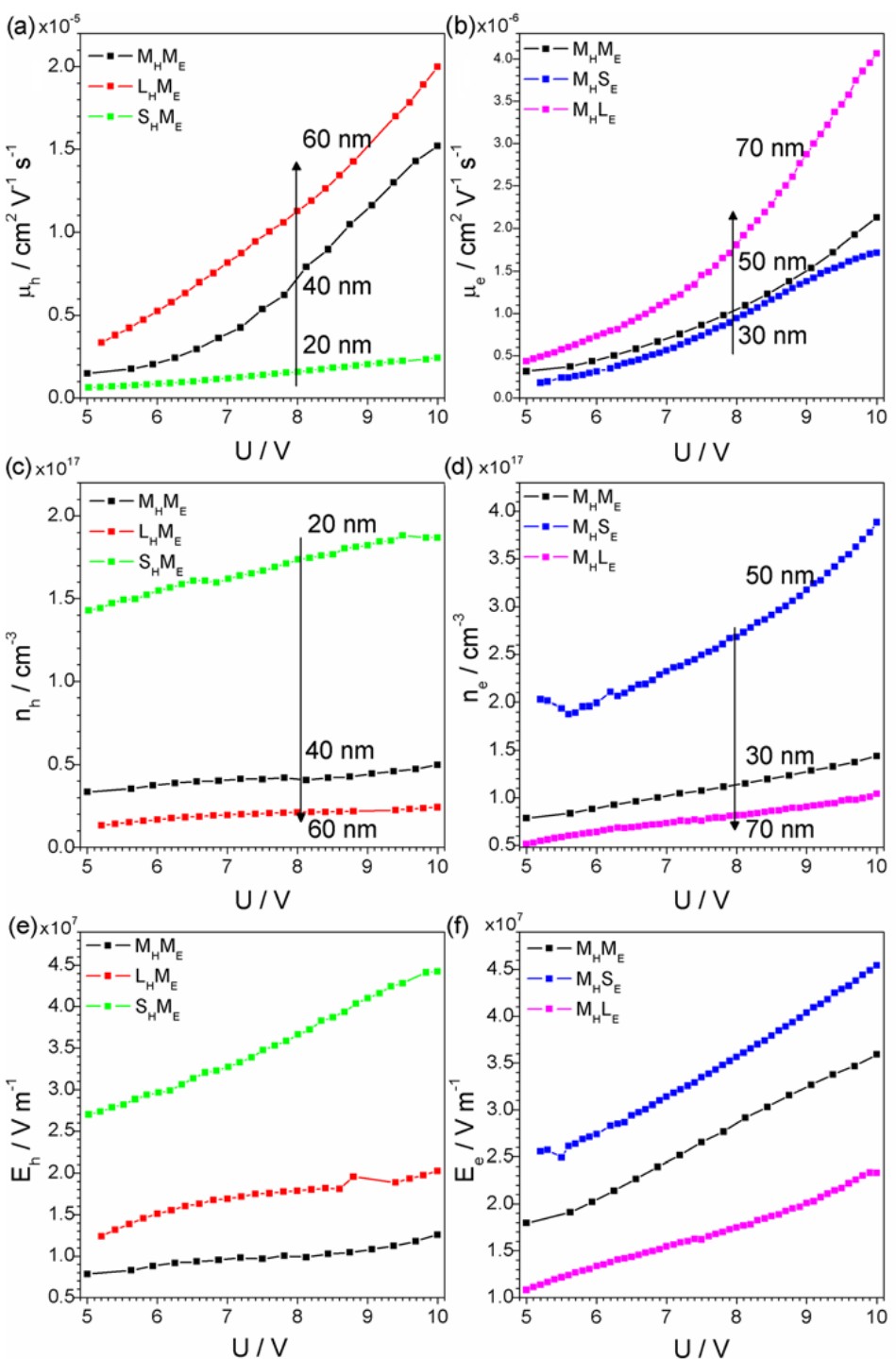

**Figure 5.** Estimated values of charge mobility (**a,b**), charge density (**c,d**) and electric field (**e,f**) in hole-transport (**a,c,e**) and electron-transport (**b,d,f**) layers as functions of the voltage applied to the whole device. Vertical arrows indicate the increase of the layer thickness.

Figure 5a,b demonstrate an inverse correlation between charge carrier mobility and layer thickness. The highest mobilities were observed for the thickest layer ($L_H M_E$ and $M_H L_E$), whereas mobilities in the thinnest layer ($S_H M_E$ and $M_H S_E$) were the lowest. An organic conductive layer is formed of bulk organic molecules (Figure 1b), so one can hardly

expect that they can conduct charges isotropically. The thickening of a layer provides more ability for molecule arrangement and assures more paths for charge transport. This results in both effects: the rise of charge mobility and the fall of the concentration of charge carriers which are weaker accumulated and better transported through the layer. A similar effect regarding mobility and thickness has already been observed for organic field-effect transistors [35,36].

The conformity of the results with the Poole–Frenkel Equation (4) was checked.

$$\mu = \mu_0 e^{\beta \sqrt{E}} \tag{4}$$

where $\mu_0$ is zero-field mobility, $\beta$ is a coefficient characterizing charge transfer activation energy.

The dependences of mobility on the electric field in Poole–Frenkel coordinates for each layer of $\mathbf{M_H M_E}$ and $\mathbf{L_H M_E}$ regarded types of devices are presented in Figure 6a,b. The data for all studied diodes is presented in Figure A2 (Appendix A). An ideal linear dependence of $\log(\mu)$ on $E^{1/2}$ could not be expected, as the arrangement of organic molecules inside the layer is non-uniform and anisotropic. Nevertheless, all the plots contain linear sections. A nearly linear dependence was observed for the ETL of $\mathbf{M_H M_E}$ (Figure 6a) and both layers of LM (Figure 6b). The Poole–Frenkel equation parameters are shown in the inset of Figure 6a,b, and Figure A2 (Appendix A). The data show a rather strong dependence of mobility on the electric field, with a $\beta$ coefficient of about $1 \times 10^{-3}$ s.

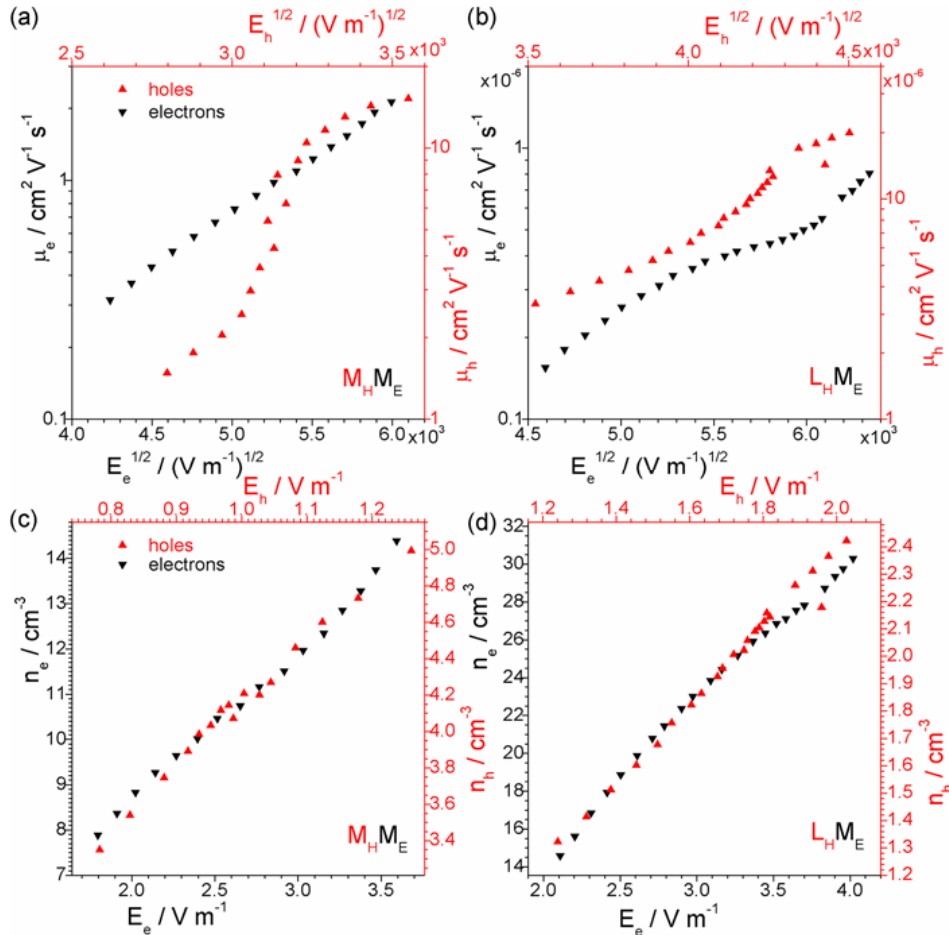

**Figure 6.** Scatter plots of charge mobility vs. electric field intensity in Poole–Frenkel coordinates (**a,b**) and charge density vs. electric field in linear coordinates (**c,d**) for $\mathbf{M_H M_E}$ (**a,c**) and $\mathbf{L_H M_E}$ (**b,d**) devices. The linearized equations with calculated $\ln(\mu_0)$ and $\beta$ values are shown in insets (**a,b**).

Another interesting correlation was elucidated by plotting the density of charge carriers vs. the electric field (Figure 6c,d, and Figure A3, Appendix A). The observed linear dependence between charge density and electric field provides an easy way to extrapolate and predict the density of charge carriers injected at the lower or higher electric fields. This extrapolation, along with extrapolation of $\log(\mu) - E^{1/2}$ (Figure 6a,b), will be considered below as a technique to predict the current–voltage behaviour of the entire multi-layer device. To continue discussion within the scope of exploitation properties, the current–voltage characteristics of the diodes have to be addressed. Even though the qualitative composition of all samples is the same, a significant difference in current–voltage plots is observed, as Figure 7 shows. Though such plots are often demonstrated in logarithmic scale, here we intend to emphasize the difference of the current for different devices, which is more clear when the linear scale is used.

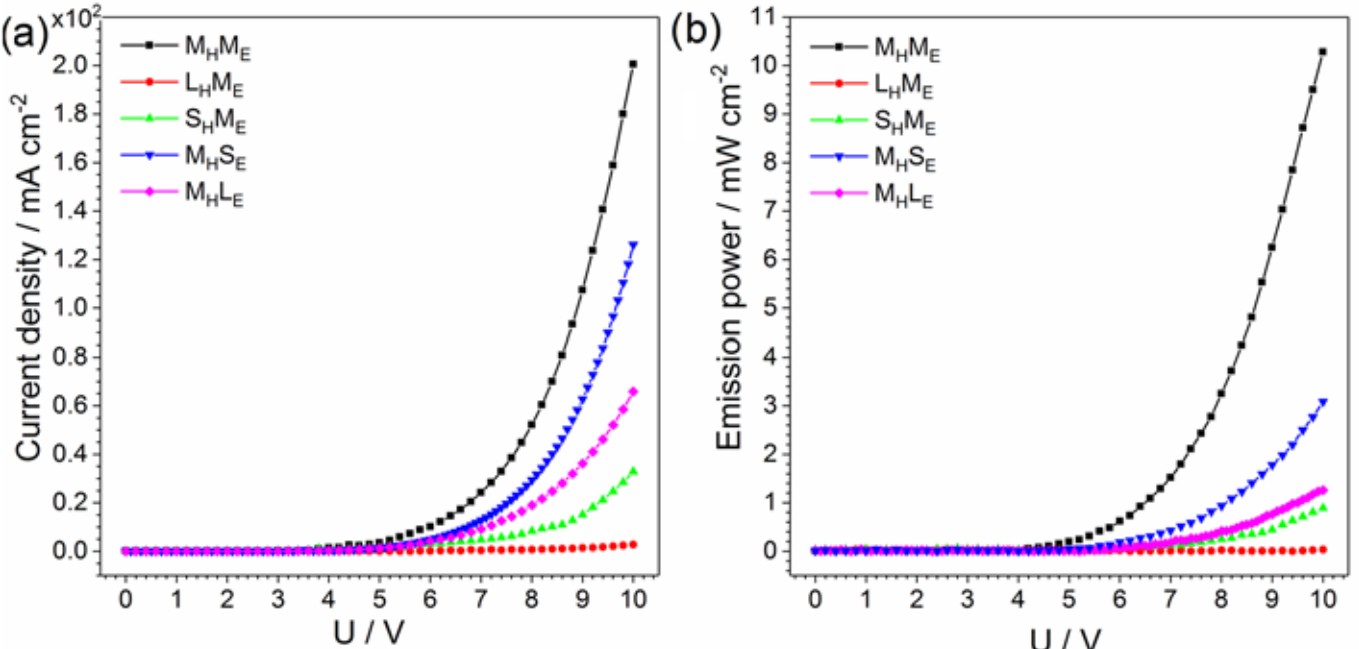

**Figure 7.** Voltammetric (**a**) and emission power (**b**) characteristics of five diodes differing by transport layer thicknesses. Average external quantum efficiency (EQE) and power efficiency are shown in the inset.

As was noted above, the factors determining current depend on layer thickness differently: a thickening of the film causes an increase in charge mobility and a decrease in charge density at the same applied voltage. The issue lies in balancing both charge density and mobility by tuning layer thickness to achieve the maximal product of those factors.

The best result has been observed in the case of MM diode, which transport layers (50 nm and 60 nm) were neither too thick nor too thin (Figure 7). The voltammetric response (Figure 7a) correlates with the electroluminance performance (Figure 7b) to a certain degree. That correlation allows for the prediction of emission properties by evaluating the current–voltage characteristic.

The elucidated linear dependencies between the parameters (Figure 6) can be used to predict the conductivity of the multi-layer diode using the schematic approach shown in Scheme 1. The predicted current density is calculated as a product of three parameters $n$, $\mu$, and $E$. The current density is always the same for all diode layers at the same time. Thus, it is useful to plot the current density value on the abscissa scale (Figure 8).

$$\lg(\mu) = f\left(\sqrt{E}\right)$$

$$I = E \times \mu \times n$$

$$n = f(E)$$

**Scheme 1.** The scheme for calculation and prediction of the *I*, *μ*, and *n* values is based on the mutual relations between them.

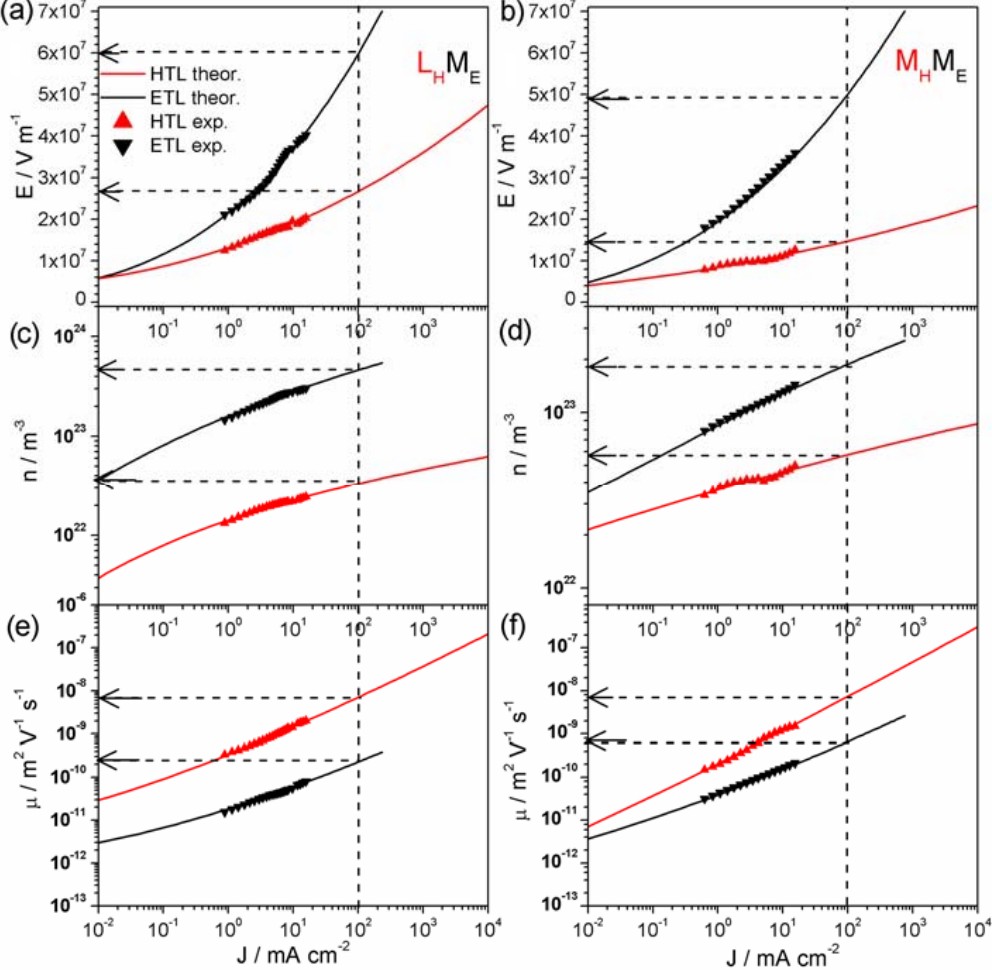

**Figure 8.** Predicting electric field (**a,b**), density (**c,d**), and charge mobility (**e,f**) in transport layers under specified working current conditions (100 mA $\times$ cm$^{-2}$) for $\mathbf{L_H M_E}$ (**a,c,e**) and $\mathbf{M_H M_E}$ (**b,d,f**) devices.

A way of predicting values of intrinsic charge transport parameters in organic layers is graphically shown in Figure 8. Two devices, $\mathbf{L_H M_E}$ and $\mathbf{M_H M_E}$ (those which showed the best and the worst results of conductivity), were chosen.

Due to technical reasons, measurement in a wide range of current is not possible, as the high current can be destructive for organic materials. Registration of impedance spectra at low current is difficult due to low signal intensity. Figure 8 demonstrates a technique for the prediction of all parameters responsible for charge transfer at an OLED operational current value of 100 mA $\times$ cm$^{-2}$. The electric field values (Figure 8a,b) are used for the evaluation

of voltage, which has to be applied to each transport layer to maintain the specified current. As Figure 8 shows, 100 mA $\times$ cm$^{-2}$ current density passing through a stack of two organic layers of $\mathbf{L_H M_E}$ requires an electric field of about $2.7 \times 10^7$ and $6.0 \times 10^7$ V·m$^{-1}$ in HTL and ETL, respectively (Figure 8a). The electric field required for the same current in the case of $\mathbf{M_H M_E}$ is noticeably smaller: $1.4 \times 10^7$ and $4.9 \times 10^7$ V $\times$ m$^{-1}$ (Figure 8b). Thus, the power efficiency of $\mathbf{M_H M_E}$ is higher than that of $\mathbf{L_H M_E}$. An analogous analysis of all five types of devices results in the same conclusion that an $\mathbf{M_H M_E}$ device would require the lowest voltage to pass the same current.

The worked-out approach can provide information for the guided search for optimal geometric parameters. The contrary effects of organic film thickness on density and mobility of injected charge carriers offer a tricky task to balance between charge density and mobility to achieve the maximal product at the minimal electric field. This task can be solved with the help of hints provided from impedance spectrum analysis.

## 4. Conclusions

A novel approach based on the analysis of impedance spectra of OLEDs has been demonstrated. A multi-step technique included the registration of a set of impedance spectra in a voltage range, the estimation of values of capacitance and resistance of diode layers, and the calculation of characteristic inherent parameters responsible for charge transfer in both hole- and electron-transport layers (charge density and charge carrier mobility). A special theoretical approach has been worked out to find a relation between experimentally estimated electric parameters (current, resistance, and capacitance) and intrinsic charge transfer kinetic parameters: charge density, charge carrier mobility, and electric field intensity. The derivation of the formulas required several assumptions. Only a single prevailing type of mobile charge carriers was regarded in a single material and the electric field was considered to be constant within the thin layer. Regardless, the assumptions appeared to be reasonable. If this were not the case, the demonstrated model could not fit the experimental data, and a much more complicated theory would have been required. Treatment of the obtained data for a set of OLEDs with a varied thickness of transport layers revealed the basic advantages and disadvantages of each type of device concerned with fundamental charge transport parameters. The effect of layer thickness on charge mobility and density is a feature that can be used to control the mentioned parameters and improve device efficiency.

**Funding:** This research was funded by Marie Sklodowska-Curie Actions within the framework programme for research and innovations Horizon 2020, project Excilight "Donor-Acceptor Light Emitting Exciplexes as Materials for Easy-to-tailor Ultra-efficient OLED Lightning" (H2020-MSCA-ITN-2015/674990).

**Institutional Review Board Statement:** Not applicable.

**Informed Consent Statement:** Not applicable.

**Data Availability Statement:** Not applicable.

**Acknowledgments:** The author acknowledges Daniel Pereira for technical support and advice on the fabrication of organic light-emitting diodes and Przemyslaw Data for his positive opinion on publishing the results.

**Conflicts of Interest:** The authors declare no conflict of interest. The funders had no role in the design of the study; in the collection, analyses, or interpretation of data; in the writing of the manuscript; or in the decision to publish the results.

## Appendix A. Results on the Characterization of the Diodes: Graphical Data

The appendix includes a set of graphical data that supplements the information presented in main text.

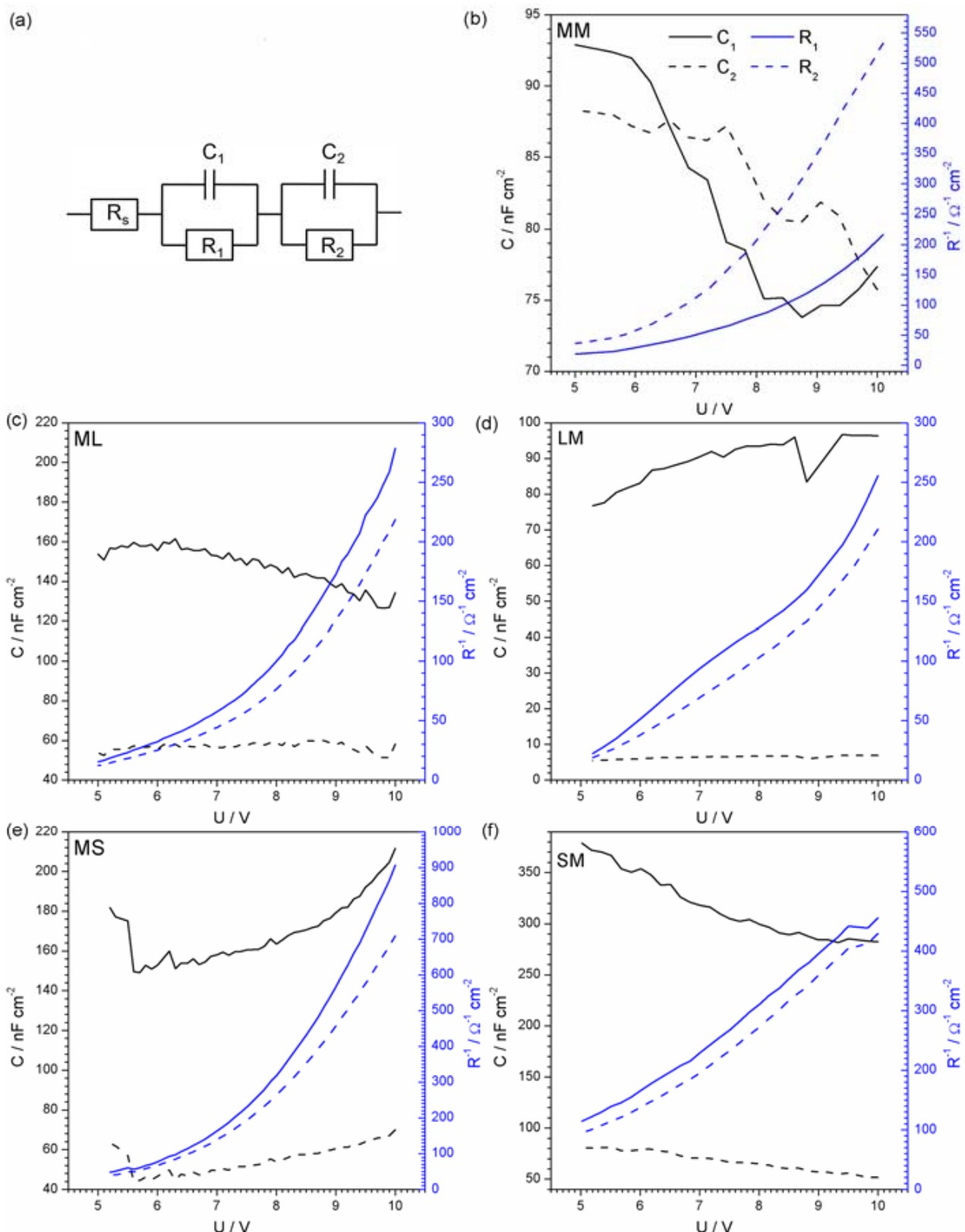

**Figure A1.** Parameters $C_1$, $C_2$, $R_1^{-1}$, $R_2^{-1}$ corresponding to equivalent electrical circuit (**a**) as functions of applied voltage for five types of studied OLEDs: MM (**b**), ML (**c**), LM (**d**), MS (**e**), SM (**f**).

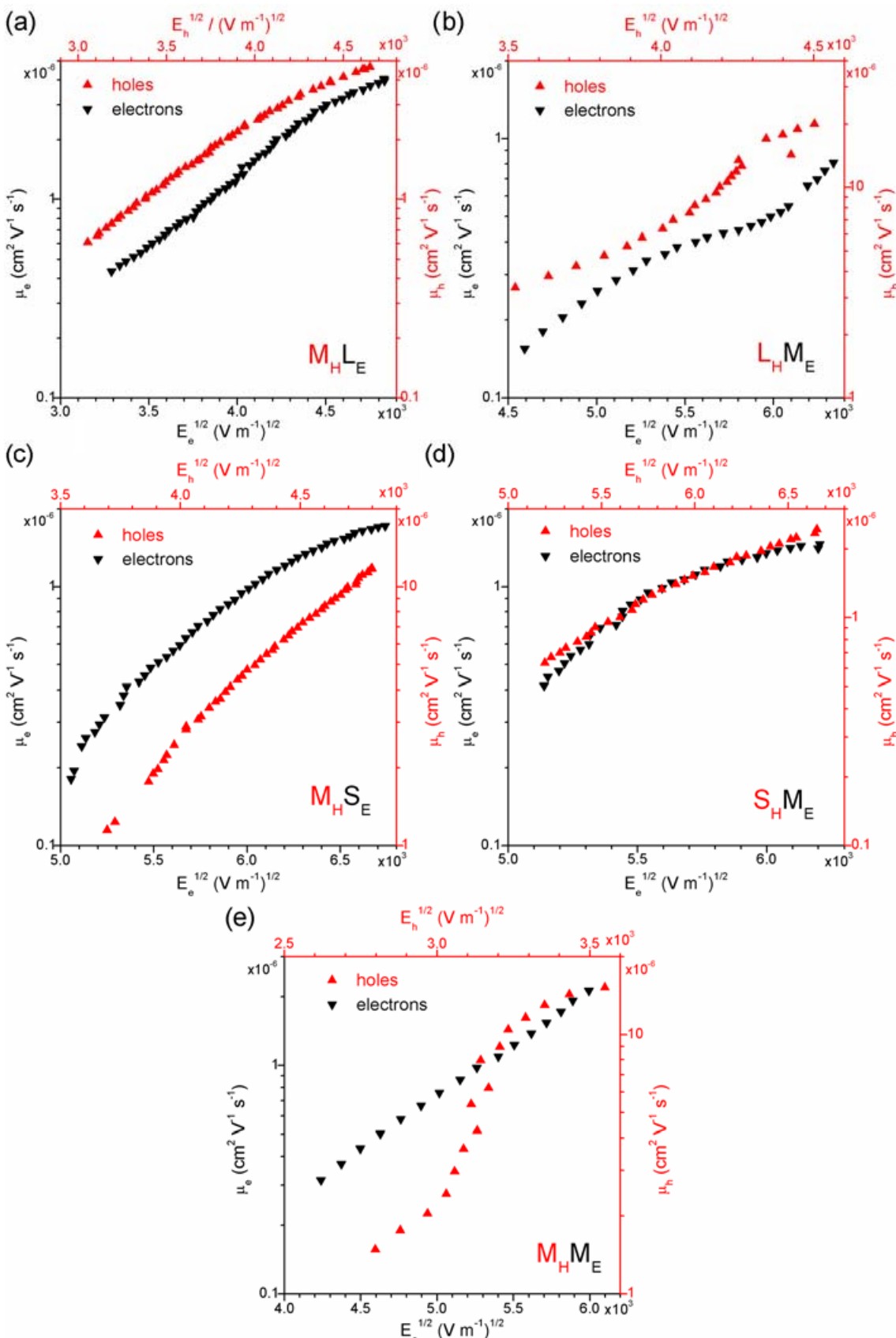

**Figure A2.** Estimated dependence scatter plots of charge mobility on the electric field in Poole–Frenkel coordinates: ML (**a**), LM (**b**), MS (**c**), SM (**d**), MM (**e**). The red and black colour of the points corresponds to holes and electrons, respectively.

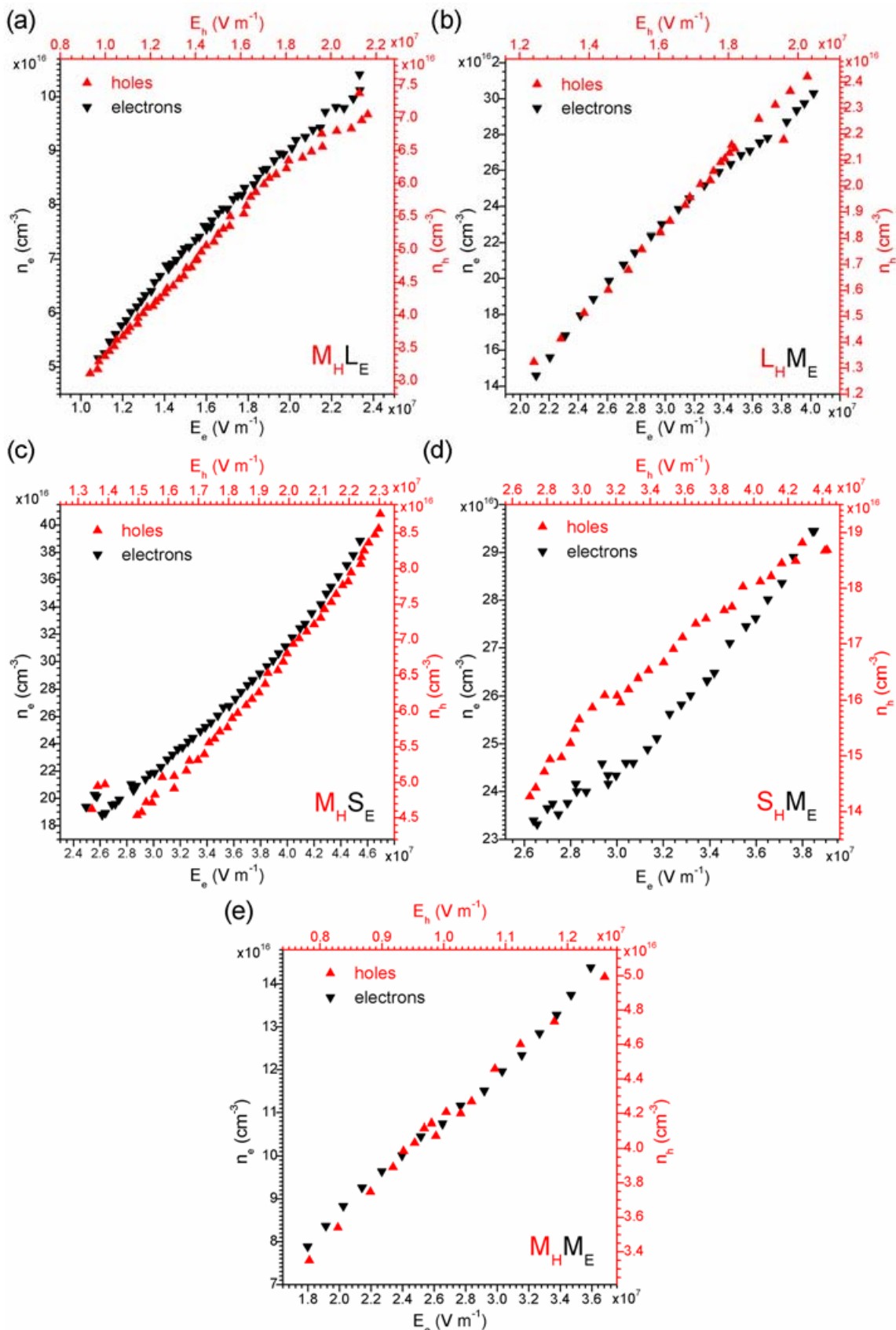

**Figure A3.** Estimated dependence scatter plots of charge density on electric field: ML (**a**), LM (**b**), MS (**c**), SM (**d**), MM (**e**). The red and black colours of data and scales correspond to holes and electrons, respectively.

## Appendix B. A Detailed Mathematical Procedure for Derivation of the Working Formulas

The text represents a single-stepping mathematical procedure that leads to the discovery of the final formula for the impedance of the organic thin layer. The designations of used constants and introduced variables along with their dimensions are listed below.

Constants:

$F$—Faraday constant (96,485 C·mol$^{-1}$);
$J$—imaginary unit ($\sqrt{-1}$);
$\varepsilon_0$—vacuum permittivity (8.85·10$^{-12}$ F·m$^{-1}$);
$\varepsilon$—relative material permittivity (−);
$R$—ideal gas constant (8.314 J·mol$^{-1}$·K$^{-1}$);
$T$—temperature (accepted to be constantly equal to 298 K);
$f$—secondary constant, equal to F/RT, introduced for simplification of expressions.

Electrochemical parameters:

$X$—distance (m);
$\Delta x$—thickness of the discrete thin layer (m);
$c$—concentration (mol·m$^{-3}$);
$\delta c$—a difference of concentration in two neighbour discrete layers (mol m$^{-3}$);
$D$—diffusion coefficient (m$^2$·s$^{-1}$);
$J$—flux (mol·m$^{-2}$·s$^{-1}$).
Electrical parameters:
$i$—current density (A·m$^{-2}$);
$U$—voltage (V);
$\delta U$—voltage drop in a discrete thin layer (V);
$E$—electric field intensity (V·m$^{-1}$);
$Z, Z_{re}, Z_{im}$—impedance (complex value), real and imaginary part ($\Omega$);
$Y$—admittance ($\Omega^{-1}$);
$C$—capacitance normalized by surface area (F m$^{-2}$);
$R$—resistance normalized by surface area ($\Omega$ m$^2$), not to be confused with molar Boltzmann constant, which is always accompanied by $T$;

Time-dependent oscillating values (dimensions are the same as for corresponding stationary values):

$\Delta i$—periodic current density oscillation;
$\Delta U$—periodic voltage oscillation;
$\Delta c$—periodic concentration oscillation.

Time-independent complex phasors:

$\widetilde{i}$—current phasor;
$\widetilde{U}$—voltage phasor;
$\widetilde{c}$—concentration phasor.

An illustration of the model depicting charge transfer through a single discrete layer is presented in Figure A4. The impedance will be firstly calculated for the one layer containing one type of charge carrier and then generalized to the whole system.

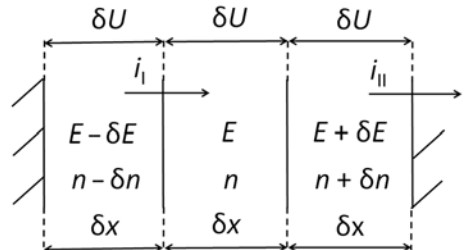

**Figure A4.** Schematic representation of three neighbor discrete layers.

To derive the final formula, one has to (i) consider oscillating parameters affecting current, (ii) represent current oscillation as a function of the other oscillating parameters, (iii) express all the parameters in terms of oscillating current and voltage, and (iv) rearrange the final formula to obtain the value of impedance.

The flux of charged species is described by Equation (A1):

$$i = -ze\frac{\mu}{f}\frac{dn}{dx} + ze\mu En + \varepsilon\varepsilon_0\frac{dE}{dt} \tag{A1}$$

The first term in (A1) is concerned with the diffusion of charged species driven by the concentration gradient. The second term refers to the migration of charge carriers due to the applied electric field. The last, third term is called displacement current, which takes place only when the electric field is changed and does not exist under DC conditions. Analysis of (A1) and comparison with experimental data declared that first and third terms were not relevant for OLED investigation under operation conditions. According to the formula (A1), the presence of displacement current must become more observable with an increase of AC frequency. However, experimental results did not allow for the observation and extraction of this factor. The diffusion (first) term seems to be negligible in comparison with the migration (second term in a strong electric field. Perhaps, further development of measurement techniques would allow the extension of the analysis by considering two omitted terms in (A1).

$$i = ze\mu En \tag{A2}$$

According to (A2) oscillating current is accompanied by the oscillation of electric field and charge density. Equations (A3) and (A4) describes current oscillation as the Taylor series of function (A2). Second and higher-order derivatives have been neglected.

$$\Delta i = \left(\frac{\partial i}{\partial E}\right)\Delta E + \left(\frac{\partial i}{\partial n}\right)\Delta n \tag{A3}$$

$$\Delta i = ze\mu n\Delta E + ze\mu E\Delta n \tag{A4}$$

The oscillating $\Delta$ values are harmonic functions of time and complex time-independent phasors (A5).

$$\Delta i = \tilde{i}\exp(j\omega t), \ \Delta n = \tilde{n}\exp(j\omega t), \ \Delta E = \tilde{E}\exp(j\omega t) \tag{A5}$$

Then, Equation (A4) gets transformed to (A6)

$$\tilde{i} = ze\mu n\tilde{E} + ze\mu E\tilde{n} \tag{A6}$$

The electric field is directly related to voltage; thus, its phasor can be substituted by a voltage phasor divided by thickness. To eliminate the charge density phasor, an additional sub-procedure has to be conducted.

For determining oscillation of charge density in the middle layer depicted in Figure A5 one has to consider the oscillation of currents $i_I$ and $i_{II}$ (A7). The formulas were obtained by adding or subtracting $\delta$ terms in Equation (A4).

$$\Delta i_I = ze\mu(n - \delta n)\Delta E + ze\mu(E - \delta E)\Delta n$$
$$\Delta i_{II} = ze\mu(n + \delta n)\Delta E + ze\mu(E + \delta E)\Delta n \tag{A7}$$

Oscillation of charge carrier concentration (charge density) must obey Fick's second law (A8),

$$\frac{\partial\Delta n}{\partial t} = -\frac{1}{ze}\frac{\partial i}{\partial x} \tag{A8}$$

which is transformed into discrete form (A9)

$$\frac{\partial\Delta n}{\partial t} = \frac{1}{ze}\frac{i_{II} - i_I}{2\delta x} \tag{A9}$$

Expansion of (A9) using (A7) leads to (A10)

$$\frac{\partial \Delta n}{\partial t} = \frac{\mu}{\delta x}(\Delta E \cdot \delta n + \Delta n \cdot \delta E) \tag{A10}$$

The last equation can then be rearranged to phasor containing form giving (A11)

$$j\omega \widetilde{n} = \frac{\mu}{\delta x}\left(\widetilde{E} \cdot \delta n + \widetilde{n} \cdot \delta E\right) \tag{A11}$$

Finally, $\widetilde{n}$ is expressed by $\widetilde{U}$ and other time-independent parameters.

$$\widetilde{n} = \frac{\mu \widetilde{E} \cdot \delta x}{j\omega(\delta x)^2 - \mu \cdot \delta E \cdot \delta x}\delta n \tag{A12}$$

The last equation provides an opportunity to replace the electric field by voltage (A13) according to (A14).

$$\widetilde{n} = \frac{\mu \widetilde{U}}{j\omega(\delta x)^2 - \mu \cdot \delta U}\delta n \tag{A13}$$

$$E = \frac{\delta U}{\delta x} \tag{A14}$$

The difference of charge density between neighbour layers ($\delta n$) will be estimated using the Poisson equation.

According to (A2), current is proportional to both electric field intensity (potential gradient) and concentration: $i \sim nE$. In a stationary state (under DC conditions), current passing through all the borders is the same; otherwise, an infinite accumulation or depletion of charge would be observed. Therefore, by equating current passing through neighbour layers (Figure A1), one can write (A15).

$$nE = (n + \delta n)(E + \delta E) \tag{A15}$$

That after rearrangement gives relation (A16):

$$\delta n = -\frac{n\delta E}{E + \delta E} \tag{A16}$$

Using the Poisson Equation (A17) in discrete form, the formula relating $\delta E$ and $n$ is obtained (A18). The Poisson equation is used in the form (A17) assuming that one charge carrier is prevailing in a single material.

$$\frac{\partial^2 U}{\partial x^2} = \frac{ze}{\varepsilon \varepsilon_0}n \tag{A17}$$

$$\delta E = \frac{ze}{\varepsilon \varepsilon_0}n \cdot \delta x \tag{A18}$$

Substituting $\delta E$ from (A18) into (A16), one obtains (A19):

$$\delta n = -\frac{e}{\varepsilon \varepsilon_0}\frac{zn^2 \cdot \delta x}{E + \frac{e}{\varepsilon \varepsilon_0}n \cdot \delta x} \tag{A19}$$

Value $E$ in the denominator must be prevailing to obey the condition $\delta n \ll n$. Thus, the formula is simplified to (A20).

$$\delta n = -\frac{E}{\varepsilon \varepsilon_0}\frac{zn^2 \cdot (\delta x)^2}{\delta U} \tag{A20}$$

Now it's time to go back to formula (A6) and substitute the values of phasors using Formulas (A13) and (A20).

$$\widetilde{i} = ze\mu n \frac{\widetilde{U}}{\delta x} - ze\mu E \boxed{\frac{\mu\widetilde{U}}{j\omega(\delta x)^2 - \mu \cdot \delta U}} \cdot \boxed{\frac{e}{\varepsilon\varepsilon_0} \frac{zn^2 \cdot (\delta x)^2}{\delta U}} \tag{A21}$$

For clarity, the fragments that have been imported from (A13) and (A20) into (A6) are marked out in the new Formula (A21)

The admittance is a ratio between oscillating current and voltage. Thus, after simplification, formulas for admittance and impedance are as follows:

$$Y = ze\mu n \frac{1}{\delta x} - \frac{z^2 e^2 \mu^2 n^2}{\varepsilon\varepsilon_0(j\omega \cdot \delta x - \mu E)} \tag{A22}$$

$$\widetilde{i} = ze\mu n \frac{\widetilde{U}}{\delta x} - ze\mu E \boxed{\frac{\mu\widetilde{U}}{j\omega(\delta x)^2 - \mu \cdot \delta U}} \cdot \boxed{\frac{e}{\varepsilon\varepsilon_0} \frac{zn^2 \cdot (\delta x)^2}{\delta U}} \tag{A23}$$

$$Y = ze\mu n \frac{1}{\delta x} + \frac{1}{\left(\varepsilon\varepsilon_0 \frac{E}{z^2 e^2 \mu n^2} - \varepsilon\varepsilon_0 \frac{j\omega \cdot \delta x}{z^2 e^2 \mu^2 n^2}\right)} \tag{A24}$$

Therefore, the impedance of the thin layer can be calculated according to Formula (A25) as the inverse admittance

$$Z = \frac{1}{ze\mu n + \frac{1}{\left(\frac{\varepsilon\varepsilon_0 E}{z^2 e^2 \mu n^2 \cdot \delta x} - j\omega \frac{\varepsilon\varepsilon_0}{z^2 e^2 \mu^2 n^2}\right)}} \delta x \tag{A25}$$

Now the formula has to be extended onto the whole layer of thickness $d$ by integrating the derivative (A26).

$$Z(d) = \int_0^d \frac{\mathrm{d}Z}{\mathrm{d}(\delta x)}\mathrm{d}(\delta x) \tag{A26}$$

The resulting formula describing a real layer would then appear in a similar form (A27).

$$Z = \frac{1}{\frac{ze\mu n}{d} + \frac{1}{\left(\frac{\varepsilon\varepsilon_0 E}{z^2 e^2 \mu n^2} - j\omega \frac{\varepsilon\varepsilon_0 d}{z^2 e^2 \mu^2 n^2}\right)}} \tag{A27}$$

The last formula does not correspond to the frequency response of a circuit that consists of capacitors and resistors. It has to be rearranged (A28) to become compatible with the traditionally used equivalent circuits including capacitors and resistors (A29).

$$Z = \frac{1}{\frac{z^2 e^2 \mu n^2}{\varepsilon\varepsilon_0 E} + \frac{ze\mu n}{d} + \frac{1}{\left(-\frac{\varepsilon\varepsilon_0 E}{z^2 e^2 \mu n^2} - j\frac{\varepsilon\varepsilon_0 E}{\omega z^2 e^2 n^2 d}\right)}} \tag{A28}$$

$$Z = \frac{1}{\frac{1}{R_\mathrm{p}} + \frac{1}{\left(R_\mathrm{s} - j\frac{1}{\omega C}\right)}} \tag{A29}$$

where

$$R_\mathrm{p} = \frac{1}{\frac{z^2 e^2 \mu n^2}{\varepsilon\varepsilon_0 E} + \frac{ze\mu n}{d}} \tag{A30}$$

$$R_\mathrm{s} = -\frac{\varepsilon\varepsilon_0 E}{z^2 e^2 \mu n^2} \tag{A31}$$

$$C = \frac{z^2 e^2 n^2 d}{\varepsilon \varepsilon_0 E^2} \tag{A32}$$

The lower indexes "s" and "p" designate parallel and series according to element position in an equivalent electric circuit (Figure A5a) which ac frequency response is described by (A29).

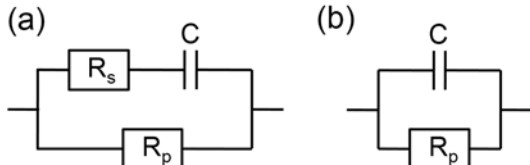

**Figure A5.** Equivalent electric circuits describing charge transfer through a thin organic layer: the complete model (**a**) and the simplified model if not very high frequencies are used (**b**).

The analysis of the data showed that the series resistance is not necessary to provide a satisfactory fit of experimental data with the theoretical model. For that reason, it could not be estimated from spectra analysis. The effect may occur due to a significant difference between the impedance of two connected in series neighbour elements with the prevailing capacitor's impedance (A33). Comparing modules of the elements, one can proceed from (A33) through (A35) and conclude that the presence of resistor $R_S$ could not be observed when condition (A35) is obeyed.

$$\frac{\varepsilon \varepsilon_0 E}{z^2 e^2 \mu n^2} << \frac{\varepsilon \varepsilon_0 E^2}{\omega z^2 E^2 n^2 \cdot \delta x} \tag{A33}$$

$$\frac{1}{\mu} << \frac{E}{\omega \cdot d} \tag{A34}$$

$$\omega << \frac{\mu E}{d} \tag{A35}$$

Since the value in the right part of the expression (A35) is large, a very high and hardly achievable frequency is required to reveal the presence of the resistor $R_S$. Otherwise, the apparent equivalent electrical circuit of the layer would contain only two elements (Figure A5b).

Two other parameters ($C$ and $R_p$) are easily estimated using spectrum analysis.

Thus, basing on experimentally measured values $i$, $C$, and $R_p$, one can calculate the values of three charge transport parameters: charge carrier mobility $\mu$ (A36), charge carrier density $n$ (A37), and electric field intensity $E$ (A38).

$$\mu = \frac{d^2}{iR^2C} \frac{1}{\left(1 + \sqrt{\frac{dC}{\varepsilon \varepsilon_0}}\right)^2} \sqrt{\frac{dC}{\varepsilon \varepsilon_0}} \tag{A36}$$

$$n = \frac{iRC}{zed}\left(1 + \sqrt{\frac{\varepsilon \varepsilon_0}{dC}}\right) \tag{A37}$$

$$E = \frac{iR}{d}\left(1 + \sqrt{\frac{dC}{\varepsilon \varepsilon_0}}\right) \tag{A38}$$

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
