# Peer review of "In-Situ Characterisation of Charge Transport in Organic Light-Emitting Diode by Impedance Spectroscopy"

_electronicmat, doi:10.3390/electronicmat2020018_

Round 1

Reviewer 1 Report

The manuscript entitled « In-situ characterization of charge transport……spectroscopy” by Chulkin et al. described the charge transport mechanism using the in-situ dielectric spectroscopy. The investigated domain of research is interesting. This paper can be accepted after the revision on several points as stated below:

  • From references 11-18, while talking about the charge transport mechanism and techniques like Time of flight, OFET, etc., authors have to include recent references with a variety of recently developed organic semiconducting molecules for the OLED devices. Authors may consult recently published papers on OLEDs: J. Mater. Chem. C, 2020, 8, 12485; Mater. Chem. C, 2019,7, 5724-5738; Light: Science & Applications, 10, 18 (2021); Mater. Chem. Front., 2020,4, 788-820; J. Mater. Chem. C, 2020,8, 2614-2642 and so on.
  • The device fabrication method needs to be elaborated; like the amount of material taken to make 1:1 ratio of NPB/TPBi, How NPB and TPBi were coated? And by which method including concentration if they were taken in solution, etc. It is also important to mention the temperature at which measurements have been taken as the performance of OLED significantly depends on the temperature. What was the humidity condition for each device?
  • In fig 2a and 2b; it is important to mention the value of voltage for every curve as the number of curves is not the same as mentioned in the experimental section (0 to 10 V increment by 0.1 or 0.25V). Authors can provide an enlarged image as supplementary.
  • In the low-frequency region, the ionic contribution is significant; have authors filtered that contribution? If yes, how they have done it. Please provide treated and untreated data as supplementary.
  • Delete the instruction sentence from the acknowledgment.

Author Response

Dear Reviewer!
Thank You for attentive review of the manuscript and giving several valuable remarks. Hereinafter, we would like to comment on Your remarks.

  1. The papers cited in the article were essential for the development of the theoretical approach or accomplishment of the data analysis. Their presence on the reference list is appropriate according to the principles of scientific publishing. After getting acquainted with the papers You recommend, we've come to the conclusion that they would not give any special idea to affect the results or their discussion. I would prefer to leave the list of cited papers without additions.
  2. The deposition technique is described in Materials&Methods section, yet it was amended according to Your remark to make the procedure description more detailed.
  3. A specification has been added to the figure caption. The values were not shown on the graph so as not to overcharge the figure with exceed of digital data. Only every second curve has been shown, i.e. with the 0.2 V increment, because dense stacking of all the spectra with 0.1 V increment was not readable.
  4. The ionic contribution refers to the case of ionic conductor (i.e. solution). In that case the lower frequency region would probably reveal presence of the diffusion impedance. This is not the case in our work. No contribution is filtered during analysis - all contributions are taken into account in the equivalent circuit. The example of untreated data are presented in graphical format in Figure 2. The treatment includes not the change of the curves, but calculation of the resistance and capacitance values that correspond to the untreated data within the simplest electrical model.
  5. The technical sentence has been deleted from the acknowledgements section. Thank You for the attention.

Reviewer 2 Report

Author report the OLED analysis by impedance spectroscopy. OLEDs are fabricated as many organic layers but the characteristics of OLEDs have been performed for overall OLED performances and the physical parameters about each organic materials. For deep understanding for OLEDs, this study will be useful. I recommend this manuscript to Electronic Materials after the minor revision.

1. Line 49-50 and Line 60

For the readers, please put symbols for some sentences in the Introduction sections like line 222.

It is a product of charge density, charge mobility. Those two mentioned parameters are functions of the electric field and depend on layer thickness.

=> It is a product of charge density (n) and charge mobility (μ). Those two mentioned parameters are functions of the electric field (E) and depend on layer thickness.

It is worth mentioning that not only charge carrier mobility, but a complex of three parameters has to be considered to predict the full conductivity characteristics of a material.

=> It is worth mentioning that not only charge carrier mobility (μ), but a complex of three parameters (n, E, μ) has to be considered to predict the full conductivity characteristics of a material.

2. Line 68-69

If possible, please rewrite this sentences. (The following is just my suggestion) In 2001 and 2003, the authors addressed only more complex effects. Later those complex effects were investigated and researched by other research group in 2011 and 2012.

The authors of the mentioned articles addressed more complex effects such as localized states [21] and shallow traps in later works [21,22].

=> The authors of the mentioned articles addressed more complex effects such as localized states and shallow traps which were investigated in later works [21,22].

3. Line 226, d – layer thickness

Define more about the layer thickness for the reader. Is this the total thickness (x+20 nm+y in Figure 1) of OLED ? Or Is this HTL or ETL thickness for hole and electron carrier, respectively? In this manuscripts, many thicknesses were mentioned. In addition, the explanation of Eh and Ee in Figure 5 and 6 should be needed to the end of Line 227.

4. Line 311 – 313 : Check lg(μ). And I recommend “Chart 1 or Scheme 1” instead of “Figure 8”

5. Line 276, 327, 329 : How about to use “x” instead of ”∙”?

Author Response

Dear Reviewer

Thank You for the opinion and useful suggestions that would definitely improve the manuscript.

Concerning the remark about thickness, several necessary adjustments have been made in the manuscript text. All the equations included the parameters referred to a layer composed of a single material, either HTL or ETL.

Several other additions and corrections in the text have been made according to Your correct advice, including the type of number format and rename of the figure to a scheme.

Reviewer 3 Report

The work of P. Chulkin deals with the investigation of OLED materials. By means of impedance spectroscopy results, a theoretical approach was used to correlate electric and intrinsic charge transfer kinetic parameters. The best multi-layers component was evidenced as a function of ETL/HTL charge carriers thicknesses.

In my opinion this article is very well presented. All results are carefully checked and discussed.

It can be accepted in the present form.

Author Response

Dear Reviewer!

Thank You for giving Your kind opinion.

Reviewer 4 Report

The authors research on the charge transport in OLED based on impedance spectroscopy and aim to predict the arrangement of fundamental layers of the device. The work is interesting and worth of study. The manuscript can be published after addressing the following questions.

  1. The authors vary the thickness of electron and hole transport layers which results in different impedance spectroscopy. Does impedance spectroscopy measurement results can fully reflect the charge transport in the layers? What about the influence of defects or some other factors?
  2. What software do the authors use to do the simulation and calculation?
  3. How do the authors fabricate the organic films for impedance spectroscopy measurement? Thermal evaporation, spin coating or drop casting?
  4. Please cite the following papers for better understanding. Nano Energy, 66,104101 (2019), Organic Electronics, 65,96-99 (2019), Physical Chemistry Chemical Physics, 21,5,2540-2546, (2019).

Author Response

Dear Reviewer!

Thank You for the remarks. In this short report all the remarks are attended. All the issues are described in the article. Yet, many of them are worth of further discussion and study.

  1. Does impedance spectroscopy measurement results can fully reflect the charge transport in the layers? - There could be a different understanding of the "full reflect". In the work, we show how to proceed from electrically measured data to the macroscopic level by estimating average values of density and mobility of the charge carriers. The next level of the study will take place at the molecular level, it would draw a more complete picture of the charge delivery between the molecules. The resulting data gives an average set of parameters concerning the whole layer, including the defects. However, there is no sense to regard the defects in the case of organic layer composed of randomly oriented large molecules. One can say, that such a system is full of defects. That's why study of electrical properties of an organic semiconductor makes a special challenge for a researcher.
  2. The software described in reference [31] has been used to fit the equivalent electrical circuit parameters to the experimental data. Its name is EISSA and it is available for free (http://www.abc.chemistry.bsu.by/vi/analyser/program/program.htm). The calculations of data massives was done in Microsoft Excel. However, there is no need to use any particular software, since every program for impedance analysis would be capable of calculating five parameters of the model. From the researcher's side, correct data interpretation and error control remain crucial.
  3. The organic films were deposited by vacuum thermal evaporation and deposition onto ITO substrates. The deposition methodology is described in Section "Materials and Methods". 
  4. We cannot agree that the proposed papers are important for the understanding of the manuscript. According to the principles of scientific publishing, the author must cite the earlier articles which made a basis for their work. The actual version of the manuscript includes the references to the works which were essential to develop the theoretical approach, to analyse the data and to make the correct conclusions.